# Efficient Topology Design for LEO Mega-Constellation Using Topological Structure Units with Heterogeneous ISLs

**DOI:** 10.3390/s25185840

**Published:** 2025-09-18

**Authors:** Wei Zhang, Tao Wu, Xucun Yan, Guixin Li, Hongbin Ma

**Affiliations:** 1Department of Electronic and Optical Engineering, Space Engineering University, Beijing 101416, China; ttc_wt98@hgd.edu.cn (T.W.); annabelleyan40@163.com (X.Y.); liguixin@hgd.edu.cn (G.L.); hongbin_ma@163.com (H.M.); 2Key Laboratory of Intelligent Space TTC&O, Ministry of Education, Beijing 101416, China

**Keywords:** mega-constellation, network topology, heterogeneous inter-satellite links, topological structure unit, multi-objective optimization

## Abstract

With the maturation of reusable launch vehicle technology and satellite mass-production capabilities, global mega-constellation projects have entered a phase of rapid expansion. Inter-satellite networking is a key approach for enhancing constellation performance, as it crucially impacts overall constellation effectiveness. However, existing studies mostly focus on the network layer protocol optimization, with insufficient attention to topological structure design, and fail to fully consider the engineering challenges associated with inter-orbit Inter-Satellite Links (ISLs). To address these issues, this paper proposes a heterogeneous ISL topology architecture for mega-constellations, centered on “stable high-speed laser backbone connection within intra-orbit planes + dynamic and flexible radio network between inter-orbit planes”. First, we clarify the optimization objectives for mega-constellation topological design under this architecture and theoretically prove that the optimization problem is NP-hard. Building on this, we introduce Topological Structure Units (TSUs) and employ a unit reuse strategy to simplify topological design. Furthermore, we propose a TSU-based heterogeneous ISL topological design algorithm. Considering the uneven satellite distribution across latitude zones within the constellation, we further propose a regional TSU-based topological design algorithm. Finally, through simulation experiments in Starlink and GW constellation scenarios, we conduct multi-dimensional verification to demonstrate the effectiveness of the proposed algorithms in reducing end-to-end delay and decreasing ISL hops.

## 1. Introduction

Enabled by the maturation of reusable launch vehicle technologies (e.g., SpaceX’s Falcon 9 facilitating low-cost frequent launches [1]) and significant enhancements in satellite modular mass-production capabilities (ability of weekly satellite production [2]), the global satellite population has experienced explosive growth [3,4]. This has propelled the development of contemporary space infrastructure at an unprecedented pace, where mega-constellation programs have become the focal point of aerospace sector competition [5,6,7,8,9]. Notable initiatives include Starlink of SpaceX [10,11], OneWeb of the UK [12], Kuiper of Amazon [13], the GW project of China [14], and the G60 network [15], aiming to deploy global satellite communication networks.

To achieve efficient data transmission and network autonomy, Inter-Satellite Link (ISL) technology has emerged [6,9,16,17]. Its key benefits include: (1) significantly reducing the number of hops for long-distance transmission, thereby lowering transmission delay; (2) substantially increasing network capacity to meet the real-time transmission requirements of massive data; and (3) enhancing the autonomous operation capability of constellations to reduce dependence on ground stations. However, Low Earth Orbit (LEO) networks face numerous networking challenges. Current studies predominantly focus on optimizing network-layer protocols, whilst overlooking the underlying key issue of topological structure design. The dynamic variation characteristics of ISLs impose new requirements for topological structure design, making topology control critical for ensuring the performance of mega-constellation networks.

In the research field of routing and network capacity for mega-constellations, current studies predominantly rely on the fundamental assumption of grid-based ISLs. Specifically, a single satellite establishes/maintains 4 ISLs, one with each of its four adjacent satellites, forming a typical “+” structure. This configuration is referred to as the “Grid-Mesh+” structure in academia [18,19]. This model has been widely adopted in numerous extant research efforts [20,21,22,23,24,25,26,27], serving as the critical foundation upon which theoretical models are constructed and algorithms are designed.

However, the assumption of the Grid-Mesh+ structure inherently imposes unnecessary limitations on satellite networking capabilities [28,29,30,31]. Given advancements in satellite technology and practical requirements, modern satellites are generally capable of establishing connections with more distant satellites beyond adjacent neighbors. The introduction of these long-distance ISLs can significantly reduce the number of hops for information transmission, greatly enhance network capacity, and substantially decrease end-to-end transmission delay. This provides a promising new avenue for optimizing the performance of mega-constellation networks.

To address the above limitations, the academic community has undertaken significant research. Wang introduced an ISL removing algorithm [28], aiming to improve average bandwidth utilization in the Grid-Mesh+ topology while ensuring network reliability and availability. Simulation experiments using the Starlink constellation demonstrate that selectively removing specific ISLs significantly enhances the bandwidth efficiency of optical satellite networks. Chen presented a simplified 3-ISL pattern derived from the Grid-Mesh+ structure [29] and derived key routing characteristics such as minimum hop count, network diameter, path cost, and average path length. Simulation data from [29] indicates that under specific conditions, certain performance indicators of this 3-ISL topology are comparable to the Grid-Mesh+ topology. Bhattacherjee proposed a “motif-based” design method [30] that optimizes constellation topology through motifs, achieving a 40% improvement in connection length and hop count performance. Han developed a non-grid mesh topology optimization algorithm (M-NGTO) based on laser ISLs [31]. Experimental results demonstrate that M-NGTO improves average performance by approximately 39.11% compared with the Grid-Mesh+ topology.

The widely adopted Grid-Mesh+ approach [18,19,20,21,22,23,24,25,26,27] provides a fixed, static backbone but lacks the flexibility to adapt to dynamic traffic patterns and does not differentiate between the heterogeneous nature of intra- and inter-orbit links. Motif-based designs [30] and ISL-removing algorithms [28] introduce a degree of modularity by optimizing sub-structures or pruning links from an over-provisioned grid. However, these methods typically assume homogeneous link capabilities and do not explicitly model the fundamental engineering distinction between stable intra-orbit laser links and dynamic inter-orbit radio links.

In engineering practice, establishing inter-orbit dynamic laser links faces significant technical hurdles, largely attributable to the current development level of laser terminal technology. Unlike intra-orbit ISLs, inter-orbit satellites experience continuous high-speed relative motion, resulting in frequent topology switching. Establishing a link between two laser terminals requires undergoing the complex Acquisition, Tracking, and Pointing (ATP) process [32,33,34]. This process is not only inherently complex but also highly susceptible to space environmental disturbances, consequently becoming a primary bottleneck limiting the large-scale application of inter-orbit laser links.

Accounting for the significant operational differences between intra-orbit and inter-orbit ISLs and practical engineering constraints, this study proposes a heterogeneous ISL architecture for mega-constellation topology design. This architecture is based on “stable high-speed laser backbone connection within intra-orbit planes + dynamic and flexible radio network between inter-orbit planes”. This design leverages the stability of relative motion within the same orbit plane to establish a stable laser communication backbone while utilizing the dynamic and flexible nature of inter-orbit radio links to accommodate continuous and high-speed relative dynamics. By doing so, it capitalizes on the complementary strengths of stable, high-throughput intra-orbital laser links and flexible, deployable inter-orbital radio links. Building upon this model, it is proven that the general topology optimization problem for mega-constellations is NP-hard. Consequently, the time complexity of optimal solution exploration grows exponentially with the increment of the constellation scale.

To address these complex challenges, this study proposes a constellation topology design method for heterogeneous ISLs based on Topological Structure Units (TSUs). First, the basic unit of heterogeneous lSL topology is clearly defined. By solving an optimization objective function, optimal TSUs are selected and replicated across all satellites to construct a high-performance overall topology. Additionally, to account for the significant non-uniform satellite distribution at different latitudes in Walker constellations, we enhance the selection mechanism for TSUs, thereby improving topological performance. The main contributions are as follows:We propose a heterogeneous ISL architecture for mega-constellation topology design, featuring a “stable high-speed laser backbone connection within intra-orbit planes + dynamic and flexible radio network between inter-orbit planes”. We also develop the corresponding mathematical model and objective function, providing a theoretical foundation for subsequent research.We prove mathematically that the heterogeneous ISL topology design problem is NP-hard. Furthermore, we demonstrate that the optimization algorithm’s time complexity exhibits exponential growth in mega-constellation scenarios, highlighting the problem’s inherent difficulty.We define TSUs and propose a heterogeneous ISL topology design algorithm based on these units. Building on this, we propose a regional-TSU-based topology design algorithm that explicitly considers the non-uniformity of satellite latitude distribution.We conduct comparative simulation experiments using the proposed algorithms for Starlink and GW constellations. Detailed results demonstrate the algorithms’ effectiveness in enhancing topological performance across multiple dimensions.

The remainder of this paper is organized as follows. Section 2 presents the models for the mega-constellation, heterogeneous ISLs, topology and network transmission, and topology design objectives, providing the theoretical foundation for the research. Section 3 analyzes the complexity of the mega-constellation topology optimization problem. Section 4 details two topology structure design algorithms, proposed based on the definition of TSUs. Section 5 verifies the performance of the proposed algorithms through simulation experiments in Starlink and GW constellation scenarios. Finally, Section 6 summarizes the main findings and outlines future research directions.

## 2. Models and Assumptions

### 2.1. Mega-Constellation Model

As a core component of the space–air–ground information infrastructure, the LEO mega-constellation network typically consists of thousands to tens of thousands of satellites, exhibiting a three-dimensional deployment feature with multi-tier orbital shells. Satellites within each orbital shell commonly adopt the regular and symmetric Walker constellation configuration [18,19,28,29,30,31], which achieves efficient construction of global seamless coverage through parameterized design. This section constructs a fundamental mega-constellation model for heterogeneous ISL analysis from two dimensions: the orbital dynamics model and constellation configuration parameters.

#### 2.1.1. Orbit Dynamics Model

The orbital motion of satellites follows Kepler’s laws of celestial mechanics, and a complete description of the orbital state requires six orbital elements [35]. For the circular orbit configuration commonly adopted by LEO mega-constellations, the orbital parameters can be simplified to a combination of orbital altitude *h* and orbital inclination *i*:Orbital altitude *h*: The orbital altitude is defined as the vertical distance from the center of mass of the satellite to the Earth’s surface. Using the Earth’s mean radius $R_{e}=6371\;\mathrm{km}$, the semi-major axis of a circular orbit can be expressed as $a=R_{e}+h$. According to Kepler’s third law, the orbital period $T_{\mathrm{orbit}}$ satisfies Torbit=2πa3a3μμ, where $\mu =3.986004418\times 10^{14}\;m^{3}/s^{2}$ is the Earth gravitational constant. Taking a typical LEO at h=550km as an example, the calculated orbital period $T_{\mathrm{orbit}}$ is 95.65 min, which directly determines the periodicity of the satellite ground track.Orbital inclination *i*: It is defined as the angle between the orbital plane and the Earth’s equatorial plane (0°≤i≤180°), reflecting the spatial orientation of the orbital plane. When i=90°, it represents a polar orbit, achieving full coverage of the polar regions; when i<90°, it is a prograde orbit, and when i>90°, it is a retrograde orbit. Considering that more than 90% of the global population resides between latitudes of ±52°, inclined orbits are often used in practical engineering to form Walker–Delta constellations (0°<i<90°, e.g., Starlink uses i=53°). This configuration ensures high-density coverage in mid-low latitudes while maintaining ISL connectivity among orbital planes.

It is important to note that satellites in LEO are subject to various orbital perturbations, including Earth’s non-spherical gravitational field (J2 and higher-order terms), atmospheric drag, solar radiation pressure, and gravitational forces from third bodies (e.g., the Sun and Moon). These perturbations cause gradual deviations from the ideal Keplerian orbits, leading to slow changes in orbital parameters over time. However, mega-constellations employ active position-keeping strategies, typically using on-board electric propulsion systems (e.g., Hall-effect thrusters), to periodically correct these deviations and maintain the intended constellation geometry and phasing. Therefore, the impact of long-term orbital perturbations on the short-term topology control cycles proposed in this work is effectively mitigated by these operational procedures.

#### 2.1.2. Constellation Configuration Parameters

This study focuses on the Walker–Delta constellation configuration with inclined orbits [35], defined by parameters N/P/F/S. The core feature is the uniform distribution of the Right Ascensions of the Ascending Nodes (RAAN) across orbital planes (the RAAN spread is 360°), forming a quasi-synchronous orbital plane array with phase differences:Total number of satellites *N*: The total number of satellites in the constellation, satisfying N=P×S, where *P* is the number of orbital planes and *S* is the number of satellites per plane.Number of orbital planes *P*: The orbital planes are uniformly distributed along the equator, with the right ascension of the ascending node separation between adjacent planes given by $\Delta \Omega_{\mathrm{RAAN}}=360{}^{\circ}/P$.Phase factor *F*: This defines the initial phase difference of satellites between adjacent orbital planes, quantified as the offset in true anomaly position between satellites in the reference plane (k=0) and a target plane. Specifically, the phase difference for the *k*-th plane is ΔΩphase=F·360°F·360°PP. This parameter determines the spatial relative position of satellites in different orbital planes, directly affecting the dynamic connection characteristics of ISLs.

### 2.2. Heterogeneous Inter-Satellite Link Model

The mega-constellation network employs a hybrid architecture utilizing both laser and radio ISLs, achieving efficient inter-satellite connectivity configuration by leveraging the complementary characteristics of the two link types. Specifically, laser ISLs primarily undertake high-speed data transmission within the same orbital plane, while radio ISLs are responsible for establishing dynamic connections across different orbital planes. This operational division stems from the significant differences in the physical properties and engineering implementations of the two link types.

#### 2.2.1. Link Characteristics Analysis

Laser ISLs employ free-space optical communication technology with advantages of high bandwidth (10–100 Gbps) and strong anti-jamming capabilities. However, there are three significant constraints [32,33,34]:Stable Alignment Requirement: A precise acquisition, tracking, and pointing (ATP) system is essential to maintain optical link alignment. The angular variation rate induced by relative satellite motion must be maintained below the system’s tracking precision (typically controlled at levels below 10 mrad/s).Link Establishment Delay: The complete ATP process (including coarse acquisition, fine alignment, and power locking) typically requires 10 to 60 s to establish a link. Furthermore, if relative orbital motion causes the pointing angle to exceed ±5 mrad, the link is prone to interruption.Poor Dynamic Adaptability: Satellites in different orbital planes can exhibit relative velocities of several kilometers per second. This results in cumulative laser beam pointing errors over time. For example, AGI Systems Tool Kit (STK) simulations indicate that at a 550 km orbital altitude, the relative angular velocity between inter-orbit satellites can exceed 20 mrad/s, making long-term stable connections difficult to achieve.

The radio links (inter-orbit connections) utilize microwave or millimeter-wave bands for communication and have two key advantages [36,37]:Rapid Link Establishment: Enabled by all-digital beamforming technology, link establishment time can be controlled within 200 ms. For instance, the Beidou-3 system achieves ISL switching in approximately 100 ms [38]. This capability supports dynamic connections between different orbit planes.Support for Technological Progress: The breakthrough of Co-time Co-frequency Full-Duplex (CCFD) technology (e.g., achieving isolation over −110 dBc via self-interference cancellation technology [37]) has significantly enhanced the spectral efficiency of radio links, providing the physical-layer capacity necessary for management of complex inter-orbit connection topologies.

This combination of rapid establishment and robust tracking performance makes radio links the ideal choice for managing the dynamic inter-orbit connections in inclined Walker–Delta constellations, which are characterized by higher relative angular velocities, particularly at high latitudes. The wider beamwidth of radio systems readily accommodates these angular variations, preventing link interruption, a challenge that would be far more difficult to overcome with laser links in the same role.

The analysis above focuses on the fundamental link existence conditions. In operational scenarios, both laser and radio ISLs are subject to spatial environmental interference that can impact link reliability and introduce failure probabilities. Laser ISLs can experience performance degradation due to solar radiation background noise (especially when pointing near the sun) and are susceptible to potential physical damage from space debris and micrometeoroids. Radio ISLs, while largely immune to traditional multipath fading in the space environment, can be affected by ionospheric scintillation, radiation belt interference, and solar radio bursts. Quantifying the failure probabilities associated with these effects and incorporating them into a comprehensive fault-tolerance analysis of the topology is a critical and necessary step for practical deployment.

#### 2.2.2. Mathematical Model of Heterogeneous ISLs

Based on the constellation configuration described in Section 2.1.2, the constellation contains *P* orbital planes, and the number of satellites in a single orbital plane is *S*. Satellites in the constellation network can be indexed by (p,l), where a satellite $s_{m}=\left(p,l\right)$ represents the *l*-th satellite in the *p*-th orbital plane. Let the position vector of satellite $s_{m}=\left(p,l\right)$ be $r_{m}\left(t\right)$. The existence condition of the ISL can be described as follows:Laser Link Model in the Same Orbital Plane: When $p_{m}=p_{n}$, the laser link exists if and only if Equation (Equation 1) is satisfied.(1)dmn(t)=rn(t)−rm(t)≤dlaser−maxω→nnmm=rn(t)−rm(t)×drn(t)dt−drm(t)dtrn(t)−rm(t)2≤ωATP−max
where $d_{mn}\left(t\right)$ is the distance between satellites $s_{m}$ and $s_{n}$, and $d_{\mathrm{laser}-\max}$ is the maximum communication distance of the laser link (with a typical maximum range of 3000 km). ${\vec{\omega }\;}_{n/m\;}$ is the relative angular velocity vector of satellite $s_{n}$ as observed from satellite $s_{m}$, and $\omega_{\mathrm{ATP}-\max}$ is the maximum tracking angular velocity of the laser ISL tracking and pointing system (as discussed previously, this is taken to be 10 mrad/s).Radio Link Model in Different Orbital Planes: When $p_{m}\neq p_{n}$, the radio link exists if Equation (Equation 2) is satisfied.(2)$$\left\{\begin{array}{c} d_{mn}\left(t\right)=∥r_{n}\left(t\right)-r_{m}\left(t\right)∥\leq d_{\mathrm{radio}-\max}\;\\ \frac{∥r_{m}\left(t\right)\times r_{n}\left(t\right)∥}{∥r_{n}\left(t\right)-r_{m}\left(t\right)∥}\geq R_{E}+h_{\min} \end{array}\right.$$
where $d_{\mathrm{radio}-\max}$ is the maximum communication distance of the radio link, typically within 6000 km, as limited by the Effective Isotropic Radiated Power (EIRP) of the antenna. Since the maximum communication distance is long, it is necessary to avoid atmospheric influence on the ISL. Therefore, the second expression in Equation (Equation 2) must be satisfied, where $R_{E}$ is the radius of the Earth, and $h_{\min}$ is the minimum propagation height above the Earth’s surface. The atmosphere above 80 km altitude contains negligible water vapor, and its influence on ISLs in the Q/V frequency band can be neglected. Therefore, $h_{\min}$ should be no less than 80 km. For example, when the orbital height *h* is the same for all satellites, it follows from Equation (Equation 2) that for satellites operating at h=550 km (typical for Starlink constellations), the maximum feasible link length $d_{\mathrm{radio}-\max}$ is given by $2\sqrt{{\left(R_{E}+h\right)}^{2}-{\left(R_{E}+h_{\min}\right)}^{2}}$, which calculates to 5013.9 km. It should be further emphasized that the maximum communication distance $d_{\mathrm{radio}-\max}$ is fundamentally constrained by the link budget. The essential link budget equation can be expressed as Equation (Equation 3).(3)$$P_{\mathrm{rx}}=P_{\mathrm{tx}}+G_{\mathrm{tx}}+G_{\mathrm{rx}}-L_{\mathrm{path}}-L_{\mathrm{atm}}-L_{\mathrm{other}}$$
where $P_{\mathrm{rx}}$ is the received power, $P_{\mathrm{tx}}$ is the transmitter power, $G_{\mathrm{tx}}$ and $G_{\mathrm{rx}}$ are the antenna gains, $L_{\mathrm{path}}$ is the free-space path loss, $L_{\mathrm{atm}}$ is the atmospheric loss, and $L_{\mathrm{other}}$ encompasses other losses (e.g., pointing loss, polarization mismatch). The achievable data rate and connection stability are highly sensitive to the choice of digital modulation and coding schemes (e.g., BPSK, QPSK, 16-QAM, 64-QAM) [39,40,41]. Higher-order modulations offer greater spectral efficiency but require a higher received Signal-to-Noise Ratio (SNR) for a target Bit Error Rate (BER) [39]. The *link margin*, defined as the excess $P_{\mathrm{rx}}$ above the minimum required for the chosen modulation, is a critical metric. A positive link margin ensures resilience against signal fluctuations. This is particularly vital during the handover process between satellites in multi-orbit networks. As a satellite moves, the link distance $d_{mn}\left(t\right)$ and path loss change dynamically. A sufficient link margin provides a buffer, preventing the SNR from dropping below the demodulation threshold before the handover is complete, thereby guaranteeing a stable and uninterrupted connection. Therefore, when designing the ISL for a specific constellation, the value of $d_{\mathrm{radio}-\max}$ must be determined not only by EIRP but also by the required modulation scheme and the necessary link margin for robust handovers.

#### 2.2.3. Analysis of Heterogeneous Link Advantages

The hybrid link architecture forms a hierarchical system consisting of “stable high-speed laser backbone connection within intra-orbit planes + dynamic and flexible radio network between inter-orbit planes”. Advantages are analyzed as follows:The ring topology within the same orbital plane, constructed by laser links (with a single-link bandwidth $B_{\mathrm{laser}}$ reaching 10–100 Gbps), performs periodic data backhauling tasks, thereby decreasing intra-orbit transmission delay.The mesh topology between different orbital planes, established by radio links (with a single-link bandwidth $B_{\mathrm{radio}}$ of 1–10 Gbps), supports dynamic routing reconfiguration and adapts to the time-varying connectivity caused by the relative motion between orbital planes (with an average link duration of approximately 5–10 min).

Through the complementary design of the two link types, the requirement for high-bandwidth backbone transmission in the LEO mega-constellation is satisfied, and the engineering challenge of dynamic inter-orbit connection is addressed via the fast setup capability of radio links. This provides reliable physical-layer support for subsequent dynamic topological structure unit (TSU) modeling.

### 2.3. Transmission Model for Mega-Constellation

Against the backdrop of continuously growing global communication demands, LEO mega-constellations emerge as a key technical solution for achieving global coverage and low-latency communication by leveraging their unique spatial advantages. Compared with terrestrial networks, mega-constellations can establish links closely approximating the geometric shortest path in space, while signals propagate at the speed of light *c* in a vacuum. This avoids the propagation delay limitation inherent in terrestrial optical fibers (where signal propagation speed is approximately 2c2c33) [42], endowing mega-constellations with remarkable potential for low latency in long-distance communication scenarios. However, transforming this potential into the practical capability to serve massive global traffic requires maximizing the utilization of the limited number and bandwidth capacity of ISLs per satellite to achieve high throughput across the constellation network.

Network topology formed by ISLs plays a decisive role in communication performance. Under the condition of limited network resources (specifically, with a fixed number of ISLs, $N_{\mathrm{ISL}}$, and fixed single-link capacity, $C_{\mathrm{ISL}}$), the hop count *H* of end-to-end connections traversing ISLs is closely related to network throughput. A connection traversing too many ISLs occupies significant link capacity resources, limiting the bandwidth available for other connections and causing overall network throughput to decline. Mathematically, consider *M* end-to-end connections in the network, where the *i*-th connection has hop count $H_{i}$ and occupies link capacity $C_{\mathrm{used},i}$. Under the constraint of link capacity, there is $\sum_{i=1}^{M}C_{used,i}\leq N_{ISL}\times C_{ISL}$, and $C_{used,i}$ is positively correlated with *H*. While accurate throughput evaluation depends on routing algorithms and traffic allocation, in the mega-constellation network with low latency as the core goal, routing inevitably favors shortest or near-shortest delay paths. Therefore, using hop count as a simplified proxy for network throughput is both reasonable and feasible. Furthermore, fewer hop counts imply fewer nodes traversed per end-to-end connection. This translates to lower resource consumption per unit of data transmitted (increased resource efficiency), enabling higher aggregate throughput for a given total network capacity or equivalent throughput with lower resource usage. Compared with metrics like network throughput, whose accurate assessment depends on complex routing schemes, hop count calculation depends solely on the network topology’s node connectivity. It abstracts away complex factors such as dynamic traffic allocation, link bandwidth variations, and detailed transmission behavior. Consequently, hop count is far more efficient to evaluate and can be readily integrated into network optimization models, providing an intuitive and efficient criterion for topology design.

In terms of delay calculation, to reduce problem complexity, this paper only considers the propagation delay of signals in ISLs, ignoring queuing and processing delays at each hop node. Let $d_{mn}$ be the distance between satellite $s_{m}$ and satellite $s_{n}$. The single-hop propagation delay is then tprop=dmndmncc, and the end-to-end propagation delay is $T_{\mathrm{end}-\mathrm{end}}=\sum_{j=1}^{h}t_{\mathrm{prop},j}=\frac{1}{c}\sum_{j=1}^{h}d_{j}$, where $d_{j}$ is the length of the *j*-th hop link.

In the mega-constellation network, end-to-end traffic demand is a key factor affecting topology design. We define the end-to-end traffic matrix $F=[f_{ij}]$, where i,j∈{1,…,L}, *L* represents the number of ground cities (e.g., set L=200), and $f_{ij}$ denotes the data traffic demand from city *i* to city *j*. This matrix characterizes the traffic distribution among ground cities and serves as an important input for optimizing the constellation topology. To simplify the problem, we make the following assumption: as long as a ground station is within a satellite’s coverage, a connection can be established between them. Furthermore, in the scenario of high-population-density cities, multiple ground stations within each city are simplified as a single site (endpoint), where the connection bandwidth and connectivity with a satellite are limited only by distance. Therefore, calculations of both the total number of ISL hops $H_{\mathrm{sum}}=\sum_{i=1}^{M}H_{i}$ and the total delay $T=\sum_{i=1}^{M}T_{\mathrm{end}-\mathrm{end},i}$ for all end-to-end connections in the network depend on the traffic demands specified by matrix F.

This modeling approach is justified for the following reasons. Although a major city may contain multiple ground stations or users, their geographic distribution is concentrated within a relatively small area compared with the coverage radius of a low-earth-orbit satellite (typically hundreds of kilometers). Consequently, from the network topology perspective, these multiple ground entities within the same city are likely to connect to the same or adjacent orbiting satellites. Therefore, aggregating them into a single node primarily simplifies the modeling of the terrestrial segment without significantly altering the analysis of the space segment (i.e., the inter-satellite multi-hop path), which is the primary focus of this study. While this simplification may not capture the finest details of ultra-dense intra-city traffic distribution, we argue that it preserves the generality of our model for evaluating overall network performance metrics, such as end-to-end latency and hop count between cities.

### 2.4. Target Model for Topology Design of Mega-Constellation

Analogizing the network routing process to vehicle driving in an urban traffic system, ISLs are akin to roads in a city, while the network topology resembles the entire urban road layout. Improper road planning leads to traffic congestion and reduces traffic efficiency; similarly, a suboptimal constellation network topology design leads to inefficient utilization of ISLs, increasing data transmission delay and hop count. Therefore, the core problem of mega-constellation topology design can be stated as follows: given the number of satellites *N*, the constraint on the number of ISLs $N_{\mathrm{ISL}}$, and the per-ISL link capacity $C_{\mathrm{ISL}}$, by optimizing the ISL connection strategy (i.e., determining the connection relationship between satellite $s_{m}$ and satellite $s_{n}$ (m≠n), which can be represented by a connection matrix $A=[a_{mn}]$, where $a_{mn}=1$ indicates a connection is established between $s_{m}$ and $s_{n}$, and $a_{mn}=0$ indicates no connection), minimize the sum of ISL hops $H_{\mathrm{sum}}=\sum_{i=1}^{M}H_{i}$ and the sum of end-to-end delays $T=\sum_{i=1}^{M}T_{\mathrm{end}-\mathrm{end},i}$ for all *M* end-to-end connections in the network, given the traffic demand represented by the traffic matrix F. This corresponds to solving the optimization problem over the connection matrix A shown in Equation (Equation 4).(4)$$\left\{\begin{array}{c} \min_{F}\;\left(\Psi =H_{sum}+\lambda T\right)\;\\ s.t.\sum_{m=1}^{N}\sum_{n=1,m\neq n}^{N}a_{mn}\leq N_{ISL}\;\\ a_{mn}\in \left\{0,1\right\},\forall m,n\in \left\{1,\ldots ,N\right\} \end{array}\right.$$

Here, Ψ is the objective function, and λ is the weight coefficient for hops and delay, used to balance their relative importance in the optimization goal. By adjusting the value of λ, the model can adapt to the differentiated requirements of network performance in different application scenarios.

## 3. Complexity Analysis of Topology Optimization Problem for Mega-Constellation

### 3.1. NP-Hard Proof of Topology Optimization Problem for Mega-Constellation

Before proving that the topology optimization problem of mega-constellation is NP-hard, we first clarify the definition of NP-hard problems: A problem is NP-hard if no deterministic algorithm can solve the problem in polynomial time, and any NP problem can be reduced to this problem in polynomial time. This means that if we can prove that the mega-constellation topology optimization problem meets the above conditions, its NP-hardness is established. We consider reducing the known NP-hard problem, the vertex cover problem [43], to the mega-constellation topology optimization problem. The vertex cover problem is described as: given an undirected graph G=(V,E) and a positive integer *k*, determine whether there exists a vertex subset S⊆V with |S|≤k such that each edge in *E* has at least one endpoint in *S*. In the mega-constellation topology optimization problem, satellites are regarded as vertices, and ISLs are regarded as edges. For the graph *G* in the vertex cover problem, a mega-constellation topology model can be constructed, where satellites correspond to the vertices of *G*, and ISLs correspond to the edges of *G*.

Finding a topology that minimizes the end-to-end connection hop count and delay requires making suitable choices about the connection relationships between satellites. This requirement resembles the task in the vertex cover problem of selecting an appropriate vertex subset *S* to cover all edges. Moreover, this transformation from the vertex cover problem to the mega-constellation topology optimization problem can be completed in polynomial time, as it only requires mapping the vertices and edges of the graph to satellites and ISLs without requiring any complex calculations. Since the vertex cover problem is NP-hard and can be reduced to the mega-constellation topology optimization problem in polynomial time, the mega-constellation topology optimization problem is also NP-hard.

### 3.2. Time Complexity Analysis of Topology Optimization Problem for Mega-Constellation

Next, we analyze the time complexity of the problem. Under ideal conditions, in the topology optimization problem of a LEO mega-constellation, the connection matrix $A=[a_{mn}]$ is used to represent the connection relationship between satellites, where $a_{mn}\in \left\{0,1\right\}$ and m,n∈{1,…,N}. For *N* satellites, the number of connection combinations is $2^{N(N-1)/2}$, since each $a_{mn}$ has two possible values (0 or 1), and the number of pairwise connections between satellites is CN2=N(N−1)N(N−1)22. When traversing all possible connection combinations to find the optimal solution using an exhaustive search method, the time complexity is $O(2^{N(N-1)/2})$. This exponential time complexity exhibits computational load growth that scales exponentially with the number of satellites *N*. The required computational time increases drastically, exceeding that of polynomial-time complexity algorithms, which further underscores the computational intractability of this problem.

The above analysis is a general analysis under simplified conditions. In the heterogeneous ISL architecture studied in this paper, laser links are dedicated to intra-orbit planes, and their link establishment rules are relatively straightforward, so we can assume that the relevant analysis can be completed within polynomial time. For the radio links used in inter-orbit planes, constrained by the orbital distribution and link establishment distance described in Section 2.2, this architecture specifies that a single satellite can only establish two radio links with $N_{c}$ surrounding satellites. Although this strict constraint reshapes the solution space structure of the topology optimization problem for mega-constellations, the time complexity remains high.

For a single satellite selecting two links among $N_{c}$ potential connection partners, the number of possible combinations is CNc2=Nc(Nc−1)Nc(Nc−1)22, which is a quadratic polynomial in terms of $N_{c}$.

Considering the entire constellation of *N* satellites and leveraging the symmetry of connection relationships between satellites, each satellite within the system must select its local topological connections. If an exhaustive search algorithm is used to find the optimal constellation topology by traversing all possible connection combinations, since the number of possible ways available to each satellite for establishing radio links is $C_{N_{c}}^{2}$, the time complexity of exhaustive search is approximately given by Equation (Equation 5). Although Nc(Nc−1)Nc(Nc−1)22 is polynomial in $N_{c}$, the exponential term *N* makes the overall time complexity still exponential.(5)O((CNc2)NN22)=O((Nc(Nc−1)2)NN22)

Especially for large-scale mega-constellations where *N* and $N_{c}$ are large, the exponential time complexity still makes it a huge challenge to obtain the optimal solution through exact solving in practical calculations, and it is difficult to complete the calculation task within an acceptable time frame.

## 4. Efficient Topology Design for LEO Mega-Constellation Based on TSUs

### 4.1. Definition of Topological Structure Unit (TSU)

#### 4.1.1. Topological Connection Pattern

In LEO satellite networks, the Grid-Mesh+ pattern is commonly adopted in the industry: a single satellite establishes four ISLs with its four adjacent surrounding satellites, forming a “+” structure [18,19]. However, extensive studies [28,29,30,31] show this pattern exhibits an efficiency bottleneck in mega-constellation scenarios, i.e., it relies solely on nearest-neighbor satellite connections and fails to fully utilize the satellites’ dense distribution (satellites could connect to more distant nodes to optimize topology).

To address this, we combine the idea of heterogeneous link configuration (Section 2.2) and propose a hybrid networking architecture using laser links and radio links, leveraging their complementary characteristics to achieve efficient inter-satellite connections. Laser links offer high bandwidth and low bit error rate, primarily carrying high-speed data transmission within an orbital plane; radio links accommodate dynamic scenarios, enabling flexible connections between different orbital planes and providing a foundation for the topology design’s dynamic expansion.

In designing the topology, for satellites within the same orbital plane, the core logic of the Grid-Mesh+ pattern is preserved: laser ISLs connect adjacent satellites, constructing a stable intra-orbit backbone transmission channel. For satellites on different orbital planes, radio links facilitate flexible topological connections based on their dynamic capability.

As shown in Figure 1, satellite $s_{m}$ has a specific radio ISL range within the mega-constellation. Should $s_{m}$ select two satellites within this range to establish links, and if other satellites (e.g., $s_{n}$) in the entire network replicate $s_{m}$’s topological connection pattern, a coherent ISL topology emerges. In this case, $s_{n}$ and $s_{m}$ share the same topological structure, enabling the definition of a reusable TSU.

#### 4.1.2. Mathematical Model of TSU

To precisely characterize the TSUs, we impose the following constraints:Link Quantity and Type Constraint: For any satellite $s_{m}$, a total of four ISLs shall be established, consisting of two intra-orbit laser links (denoted as $L_{m-o}$, where o∈{1,2} represents the two adjacent links within the same orbit plane) and two inter-orbit radio links (denoted as $R_{m-p}$, where p∈{1,2} represents the dynamic inter-orbit links). This is mathematically expressed as Equation (Equation 6).(6)$$\left\{\begin{array}{c} \left|\left\{L_{m-o}|o=1,2\right\}\right|=2\\ \left|\left\{R_{m-p}|p=1,2\right\}\right|=2 \end{array}\right.$$Intra-orbit Link Fixity: As illustrated in Figure 1, for any satellite $s_{m}$, a fixed ISL shall be established with its adjacent satellite $s_{m+1}$ in the orbital forward direction within the same orbit plane. This implies that if the adjacent satellites of $s_{m}$ are $s_{m-1}$ and $s_{m+1}$, then $L_{m+1}=\left(s_{m},s_{m+1}\right)$, where $\left(\cdot ,\cdot \right)$ denotes the ISL relationship between two satellites. Similarly, for satellite $s_{m-1}$, the ISL with its adjacent satellite $s_{m}$ in the orbital forward direction is given by $L_{m}=\left(s_{m-1},s_{m}\right)$.Inter-orbit Link Selection: Satellite $s_{m}$ shall select one satellite (denoted as $s_{m-sel}$, where $s_{m-sel}\in S_{m-reach}$) from its set of reachable satellites $S_{m-reach}$ (containing all inter-orbit satellites that $s_{m}$ can establish links with) to form a radio link $R_{m-sel}=\left(s_{m},s_{m-sel}\right)$. This selection pattern is repeated across the entire constellation. That is, for any satellite $s_{k}$ satisfying the topological reuse conditions, $s_{k}$ shall also select a corresponding satellite from its set of reachable satellites $S_{k-reach}$ to establish a link.Topological Structure Mutual Completeness: As shown in Figure 1, the remaining laser link of satellite $s_{m}$ shall be established with another satellite as defined in Constraint 2. This implies that each satellite $s_{m}$ is responsible for actively establishing one fixed intra-orbit ISL with its adjacent satellite $s_{m+1}$ in the orbit forward direction. This actively established link is denoted as $L_{m+1}=\left(s_{m},s_{m+1}\right)$. Consequently, the other intra-orbit link to its backward neighbor $s_{m-1}$, denoted $L_{m}=\left(s_{m-1},s_{m}\right)$, is not established by $s_{m}$ itself but is passively received as the result of satellite $s_{m-1}$ performing its own active connection (i.e., establishing $L_{m}=\left(s_{m-1},s_{m}\right)$).The remaining radio link (denoted as $R_{m-pass}$) is determined by another satellite via the active selection process in Constraint 3. Specifically, there exists a satellite $s_{j}$ such that it actively selects $s_{m}$ to form $R_{j-sel}=\left(s_{j},s_{m}\right)$, which, from the perspective of $s_{m}$, is its passive radio link $R_{m-pass}$. Specifically, there exists a satellite $s_{j}$ such that $R_{m-pass}=\left(s_{j},s_{m}\right)$, and the selection of $s_{m}$ by $s_{j}$ follows the rules specified in Constraint 3.

With the above constraints, the local connection view of each satellite is fully consistent. The ISL selection mode of satellite $s_{m}$ is defined as a “TSU”. According to the principle of topological structure mutual completeness (as defined in Constraint 4), each TSU is constrained to contain only one node and two links, as shown in Equation (Equation 7), where TSU(·) represents the topological structure unit of the satellite.(7)$$TSU\left(s_{m}\right)=\left\{s_{m};L_{m+1},R_{m-sel}\right\}$$

It is worth noting that the traditional Grid-Mesh+ mode can be regarded as a special case of the TSU under this definition (where inter-orbit link selection and reciprocity rules are relatively simple and fixed). On the basis of maintaining the stability of the backbone links in the same orbit plane, this method can exhaustively list all possible TSUs through constraint expansion, providing a wider range of candidate schemes for the topology design of mega-constellations.

#### 4.1.3. Universality Guarantee of TSUs

Due to the spatial distribution of satellites across orbital planes, spacing is larger near the equator and smaller at higher latitudes. Consequently, equatorial satellites have the smallest number of satellites within their reachable range at any given time. To ensure universality (meaning the links within each defined TSU are feasible for any satellite in the constellation), we select satellites near the equator at time *t* as the reference benchmark for unit design. Thus, topological connection rules defined using the equatorial satellite reachable range criteria inherently satisfy the link establishment conditions in the denser high-latitude regions. This guarantees the effectiveness and consistency of the TSUs throughout the constellation.

#### 4.1.4. Properties of TSUs

Topological Connectivity StabilityAs the fundamental unit for constructing the topology of a mega-constellation, the TSU covers only a finite subset of the ISL topological space, yet it uniquely characterizes the stable connectivity of structured topologies. As previously defined in Section 2.3, the connection matrix is $A=[a_{mn}]$, where $a_{mn}=1$ indicates a connection between satellites $s_{m}$ and $s_{n}$, and $a_{mn}=0$ indicates no connection (m≠n). If satellite $s_{m}$’s TSU includes connections to satellites $s_{a}$ and $s_{b}$ (encompassing both intra-orbit laser links and inter-orbit radio links), then for any two time points $t_{1},t_{2}\in \left[t_{0},t_{0}+T\right]$ within an observation duration *T* during the constellation’s operational period, the connection matrix elements $a_{ma}$ and $a_{mb}$ remain constant, satisfying Equation (Equation 8).(8)$$\left\{\begin{array}{c} a_{ma}\left(t_{1}\right)=a_{ma}\left(t_{2}\right)\\ a_{mb}\left(t_{1}\right)=a_{mb}\left(t_{2}\right) \end{array}\right.$$This indicates that the connection relationships within the TSU do not change with time. This time invariance of the matrix elements demonstrates the connectivity stability inherent in the TSU.Synchronized Movement and Relative Velocity CharacteristicsAs shown in Figure 1, the connections between satellite $s_{m}$ and its surrounding satellites within a TSU demonstrate synchronized movement behavior. Represented by the connection matrix A, if satellite $s_{m}$ is connected to satellite $s_{k}$ in a different orbital plane at time *t* (i.e., $a_{mk}\left(t\right)=1$), it follows from the orbital motion of the satellites that $a_{mk}\left(t+\Delta t\right)=1$ at time t+Δt. This reflects the connection’s synchronized movement. From the perspective of satellite orbit dynamics, the orbital motion of satellite $s_{m}$ and satellite $s_{k}$ within the same TSU satisfies a movement synchronization constraint. For connections spanning different orbital planes, the distance between orbital planes is larger near the equator, resulting in a smaller relative angular velocity Δω between satellites. Conversely, in higher-latitude regions, the distance decreases, leading to an increased relative angular velocity Δω. As the traditional Grid-Mesh+ mode can be regarded as a special case of the TSU proposed in this study, a similar law applies: connections between satellites in different orbital planes also exhibit smaller relative angular velocities near the equator and larger ones at high latitudes due to orbital geometric characteristics. Expressed via the connection matrix, the pattern of relative angular velocity variation corresponding to the time-varying connection between satellites $s_{p}$ and $s_{q}$ in different orbital planes under the Grid-Mesh+ mode is consistent with that of satellites $s_{m}$ and $s_{k}$ under the TSU model.Mathematical Characterization of Stable ConnectionsBased on the connection matrix A and the above dynamic characteristics, the long-term stable connection properties of the TSU are presented. Due to the consistent relative motion law between satellites and the time invariance of the connection relationship within the TSU, satisfying the time invariance of $a_{mn}\left(t\right)$, for any satellite $s_{m}$ and its connected satellite set within the topological structure unit $S_{\mathrm{TSU}}\left(s_{m}\right)=\left\{s_{m1},s_{m2},s_{m3}\right\}$, within the constellation operation period $T_{\mathrm{Orbit}}$, the row elements of the connection matrix at time *t* and t+Δt ($\Delta t\leq T_{\mathrm{Orbit}}$) satisfy Equation (Equation 9).(9)$$\left\{a_{mk}|s_{k}\in S_{\mathrm{TSU}}\left(s_{m}\right),t\right\}=\left\{a_{mk}|s_{k}\in S_{\mathrm{TSU}}\left(s_{m}\right),t+\Delta t\right\}$$That is, each satellite always maintains connections with the same satellites. Even if satellites in high-latitude regions change their operation directions, the connection relationships remain unchanged based on the connection constraints of the TSU. From the perspective of topological stability theory, such long-term stable connections enable the TSU to provide time-invariant connections within the constellation topology. We define the stable connection duration $T_{\mathrm{stable}}$ of the TSU. Since the elements of the connection matrix are time-invariant and the relative motion of satellites does not disrupt the connections, $T_{\mathrm{stable}}$ equals the operation period of the constellation.

### 4.2. Topology Design Based on TSUs

#### 4.2.1. Using a Unified TSU Across the Constellation

The following systematically presents the Unified Topological Structure Unit Algorithm (Unified TSU Algorithm) for mega-constellations using a unified TSU across the entire constellation.
Determine the reference satellite: At time *t*, calculate the vertical distance of all satellites in the constellation from the equatorial plane, and select the satellite $s_{m}$ with the smallest distance to the equator as the reference satellite. Let $d_{i}$ be the vertical distance of satellite $s_{i}$ from the equatorial plane; then $m=\arg\min_{i}\left(\right|d_{i}\left|\right)$, where *i* iterates over all satellites in the constellation.Initialize the number of ISLs: For the selected reference satellite $s_{m}$, determine that the total number of ISLs to be established is four, including two intra-orbit laser links and two inter-orbit radio links.Establish intra-orbit laser links: For satellite $s_{m}$, within its orbit plane, establish a fixed inter-satellite laser link $L_{m+1}=\left(s_{m},s_{m+1}\right)$ with the adjacent satellite $s_{m+1}$ in the forward direction. In the satellite orbit operation model, the adjacent satellite $s_{m+1}$ can be uniquely determined based on orbital parameters and satellite operation rules. This link remains fixed in subsequent topological structures.Select inter-orbit satellites to establish radio links: First, determine the set of reachable satellites for radio ISLs from satellite $s_{m}$, denoted as $S_{m-reach}$. This set includes all inter-orbit satellites with which $s_{m}$ can establish links. Select one inter-orbit satellite from $S_{m-reach}$, denoted as $s_{m-sel}$, and establish a radio link $R_{m-sel}=\left(s_{m},s_{m-sel}\right)$.Construct the TSU: Based on the established links, define a topological structure unit $TSU\left(s_{m}\right)=\left\{s_{m};L_{m+1},R_{m-sel}\right\}$, which specifies the local topological connection relationship with satellite $s_{m}$ as the core.Search for solutions of the TSU: For each satellite in the reachable satellite set $S_{m-reach}$, attempt to establish a radio link according to the method in Step 4, thereby obtaining different solutions for the TSU. Since only one satellite needs to be selected from the set to establish the link, the solution space is small and the algorithm can converge quickly.Determine the remaining link connections: The remaining intra-orbit laser link for satellite $s_{m}$ (in the backward direction) is actively established by the adjacent satellite within its orbit plane, conforming to the $TSU(s_{m})$ configuration. Similarly, the remaining radio link (denoted as $R_{m-pass}$) is actively established by another satellite, also conforming to the $TSU(s_{m})$ configuration.Apply the TSU: Perform the same TSU construction process for all satellites $s_{i}$ in the constellation to ensure the entire constellation uses a unified TSU for topology construction.Evaluate the TSU performance: Using the objective function $\Psi =H_{sum}+\lambda T$ (detailed in Section 2.4), evaluate the performance of the constellation topology constructed using a unified TSU under the traffic matrix F.Iteratively optimize the TSU: Loop through Steps 7 to 9, continuously adjust the link connections within the TSU, calculate the corresponding optimization objective function value Ψ, and find the TSU that minimizes Ψ. The entire network topology formed by this TSU is the final topology design result.

We present Algorithm 1 to provide an intuitive illustration of the Unified TSU Algorithm. Notably, the iterative optimization loop in the presented algorithm terminates upon a natural completion condition: the exhaustive evaluation of all candidate topological structure units in the solution set TSU. This set is finite and is generated by considering all possible inter-orbit link selections from the reachable satellite set $S_{m-reach}$ for the reference satellite. The algorithm iterates through each candidate TSU in TSU, constructs the corresponding global topology, computes the performance metric Ψ, and ultimately selects the configuration that minimizes Ψ. No additional heuristic stopping criteria (e.g., based on iteration count or convergence threshold) are required, as the solution space is discrete and enumerable for a given instant in time.
**Algorithm 1** Unified TSU Algorithm.**Input:** Satellite constellation *S*, traffic matrix F, weight coefficient λ**Output:**
Optimal network topology $T_{\mathrm{opt}}$
1:Select the satellite $s_{m}$ closest to the equator at time *t*2:Initialize the solution set of topology structure units TSU=⌀3:**for** each satellite $s_{i}\in S$ **do**4:   Establish an ISL $L_{i+1}=\left(s_{i},s_{i+1}\right)$ with the adjacent satellite in the forward direction of the same orbital plane5:   Select an inter-orbit satellite $s_{i-sel}$ from the reachable satellite set $S_{i-reach}$6:   Establish a radio link $R_{i-sel}=\left(s_{i},s_{i-sel}\right)$7:   Construct the topology structure unit $TSU\left(s_{m}\right)=\left\{s_{m};L_{m+1},R_{m-sel}\right\}$8:   Add $TSU(s_{i})$ to the solution set TSU9:**end for**10:Initialize the optimal objective value $\Psi_{\mathrm{opt}}=+\infty$11:Initialize the optimal topology structure $T_{\mathrm{opt}}=⌀$12:**for** each topology structure unit TSU∈TSU **do**13:   Construct the entire network topology T based on TSU14:   Calculate the total hop count $H_{\mathrm{sum}}=\sum_{i=1}^{M}H_{i}$ under the constraints of the traffic matrix F15:   Calculate the total delay $T=\sum_{i=1}^{M}T_{\mathrm{end}-\mathrm{end},i}$ under the constraints of the traffic matrix F16:   Calculate the optimization objective value $\Psi =H_{\mathrm{sum}}+\lambda T$17:   **if** $\Psi <\Psi_{\mathrm{opt}}$ **then**18:     $\Psi_{\mathrm{opt}}\leftarrow \Psi$19:     $T_{\mathrm{opt}}\leftarrow T$20:   **end if**21:**end for**22:Return the optimal topology $T_{\mathrm{opt}}$


#### 4.2.2. Topology Design Based on Regional TSUs

In Section 4.2.1, we applied a unified TSU to the entire constellation for preliminary topology design. However, satellite distribution in a typical Walker constellation exhibits significant non-uniformity across latitudes. According to orbital distribution, a Walker constellation with height *h* and inclination *i* exhibits continuous variation in subsatellite point latitude during operation. At higher latitudes, the number of elements in the reachable satellite set $S_{m-\mathrm{reach}}$ (i.e., all inter-orbit satellites that can establish links with satellite $s_{m}$) significantly exceeds that near the equator. Consequently, when satellite $s_{m}$ traverses high-latitude regions, the topological structure units $TSU\left(s_{m}\right)=\left\{s_{m};L_{m+1},R_{m-sel}\right\}$ exhibit greater abundance and diversity.

To fully utilize this characteristic, and considering that the topology of a Walker mega-constellation is essentially a torus topological structure, satellite nodes continuously experience ascending (moving to higher latitudes) and descending (moving to lower latitudes) orbital processes during operation. Therefore, in the topology design, the ascending and descending characteristics of satellite orbital operation must be taken into account.

For a Walker constellation with an orbit height of *h* and an inclination of *i*, its orbital period is $T_{\mathrm{orbit}}$. According to the satellite orbital motion law, the orbital period is uniformly divided into four quadrants, as shown in Figure 2. Among them, the second and fourth quadrants correspond to the ascending phase of the satellite, and the first and third quadrants correspond to the descending phase. Taking the second quadrant of the ascending phase as an example, during the satellite’s operation in this period, the subsatellite point latitude continuously changes from 0° to latitude *i* (the same numerical value as the orbital inclination).

To achieve more precise control over the constellation topology, we trisect the time in the second quadrant, with each segment having a duration of $\Delta T=T_{\mathrm{orbit}}/12$. Based on this time division and combined with the latitude variation of the subsatellite point, the satellite operation area can be divided into three latitude zones. Taking a Walker constellation with an orbital altitude h=550km and an inclination i=53° as an example, its orbital period is $T_{\mathrm{orbit}}=5738.99\;s$ (i.e., 95 min and 38.99 s), so $\Delta T=T_{\mathrm{orbit}}/12=478.25\;s$ (i.e., 7 min and 58.25 s).

Figure 2 shows a front projection view of the orbit, but the actual satellite orbit has an orbital inclination of 53°, from which three latitude zones can be delimited: 0°∼23.8° (1N), 23.8°∼44.1° (2N), and 44.1°∼53° (3N). Considering the symmetry of the northern and southern hemispheres, the southern hemisphere also has three corresponding latitude zones: 0°∼−23.8° (1S), −23.8°∼−44.1° (2S), and −44.1°∼−53° (3S). Since the latitude zones are divided according to the orbital period, the satellite has the same residence time in different latitude zones within a quadrant of the orbital period.

Since the topological structure of the mega-constellation is a torus dynamic topology, satellite nodes continuously experience orbit ascending and descending. If different TSUs generated in the corresponding regions are respectively adopted in the three latitude zones, for the descending orbits case, the inter-satellite distance will change during the process of crossing different latitude regions, and the TSUs generated in high-latitude regions may fail because the inter-satellite distance exceeds the effective link range.

To solve the above problem, we designate 2N and 2S as transition zones. The specific strategy is as follows: the TSUs generated based on the satellite of the lowest latitude in 1N are adopted in regions 1N, 1S, 2N, and 2S; the TSUs generated based on the satellite of the lowest latitude in 2N are adopted in regions 3N and 3S. In this way, the topological update period of the entire network is consistent with the latitude zone division time, that is, $\Delta T=T_{\mathrm{orbit}}/12$.

The division is based on the fundamental principle of ensuring that the satellite has an equal residence time ($\Delta T=T_{\mathrm{orbit}}/12$) in each of the three zones during its pass through the quadrant. The specific latitude boundaries are therefore calculated based on the orbit inclination *i* and the actual on-orbit operation of the satellite, and specifically by mapping these equal time intervals to the corresponding latitude range traversed by the satellite. This makes the method universally applicable to Walker–Delta constellations of any inclination.

The specific implementation details are shown in Algorithm 2. First, a reference satellite $s_{m}$ with the lowest latitude is selected from Region 1N. Similar to Algorithm 1, an optimal topological structure unit $TSU_{\mathrm{opt}1}$ that minimizes the network-wide optimization objective Ψ is selected. Then, the topological structure unit $TSU_{\mathrm{opt}1}$ is applied to satellites whose subsatellite points are in Regions 1N, 2N, 1S, and 2S. Next, a reference satellite $s_{n}$ with the lowest latitude is selected from Region 2N, and the topological structure unit $TSU_{2}$ generated by $s_{n}$ is applied to satellites whose subsatellite points are in Regions 3N and 3S. After this screening step, an optimal topological structure unit $TSU_{\mathrm{opt}2}$ that minimizes Ψ is obtained. Finally, the network-wide topology constructed by both $TSU_{\mathrm{opt}1}$ and $TSU_{\mathrm{opt}2}$ is established.

The execution flow of Algorithm 2 (Regional TSU Algorithm) is similar to that of Algorithm 1 described in Section 4.2.1. However, it introduces a latitude band region division mechanism as the basis for dynamic programming of the network-wide topological structure. This algorithm enables dynamic optimization in the topology control strategy, shifting from static planning. In contrast to the fixed topology design of Algorithm 1, Algorithm 2 employs a periodic dynamic topology control mechanism with a planning period $\Delta T=T_{\mathrm{orbit}}/12$. Within each planning period ΔT, the regional TSUs reconstruct the topology based on the current network traffic matrix $F=[f_{ij}]$.
**Algorithm 2** Regional TSU Algorithm.**Input:** Satellite constellation *S*, traffic matrix F, weight coefficient λ**Output:** Optimal network topology $T_{\mathrm{opt}}$
1:Divide the orbit period $T_{\mathrm{orbit}}$ into 4 quadrants: ascending in quadrants 2 and 4, descending in quadrants 1 and 32:Divide the second quadrant (latitude: 0° to *i*) into 3 equal time intervals of $T_{\mathrm{orbit}}/12$3:Define 6 latitude zones: 1N, 2N, 3N, 1S, 2S adn 3S based on the divided time intervals4:**Phase 1: Generate Optimal TSU 1**5:Select reference satellite $s_{m}$ closest to the equator in zone 1N at time *t*6:Initialize ISLs count for $s_{m}$: 2 laser links and 2 radio links7:Establish forward intra-orbit laser link $L_{m+1}=\left(s_{m},s_{m+1}\right)$8:Determine reachable inter-orbit satellites $S_{m-reach}$9:Initialize solution set ${\mathrm{TSU}}_{1}=⌀$10:**for** each satellite $s_{j}\in S_{m-reach}$ **do**11:   Establish inter-orbit radio link $R_{m-sel}=\left(s_{m},s_{j}\right)$12:   Construct topology structure unit $TSU_{1}\left(s_{m}\right)=\left\{s_{m};L_{m+1},R_{m-sel}\right\}$13:   Add $TSU_{1}\left(s_{m}\right)$ to ${\mathrm{TSU}}_{1}$14:**end for**15:Initialize $\Psi_{\mathrm{opt}1}=+\infty$ and $TSU_{\mathrm{opt}1}=⌀$16:**for** each $TSU_{1}\in {\mathrm{TSU}}_{1}$ **do**17:   Construct the entire network topology $T_{1}$ based on $TSU_{1}$18:   Compute $H_{\mathrm{sum}}$ and *T* for $T_{1}$ under the constraints of the traffic matrix F19:   Calculate $\Psi_{1}=H_{\mathrm{sum}}+\lambda T$20:   **if** $\Psi_{1}<\Psi_{\mathrm{opt}1}$ **then**21:     $\Psi_{\mathrm{opt}1}\leftarrow \Psi_{1}$22:     $TSU_{\mathrm{opt}1}\leftarrow TSU_{1}$23:   **end if**24:**end for**25:**Phase 2: Generate Optimal TSU 2 and Return the Optimal Topology**26:Initialize the optimal network topology $T_{\mathrm{opt}}=⌀$27:Select reference satellite $s_{n}$ closest to the equator in zone 2N at time *t*28:Repeat steps 6-14 for $s_{n}$ to obtain ${\mathrm{TSU}}_{2}$29:Apply $TSU_{\mathrm{opt}1}$ to satellites in zones: 1N, 2N, 1S and 2S30:**for** each $TSU_{2}\in {\mathrm{TSU}}_{2}$ **do**31:   Construct the entire network topology $T_{2}$ by applying $TSU_{2}$ to the satellites in zones 3N and 3S32:   Compute $H_{\mathrm{sum}}$ and *T* for $T_{2}$ under the constraints of the traffic matrix F33:   Calculate $\Psi_{2}=H_{\mathrm{sum}}+\lambda T$34:   **if** $\Psi_{2}<\Psi_{\mathrm{opt}2}$ **then**35:     $\Psi_{\mathrm{opt}2}\leftarrow \Psi_{2}$36:     $TSU_{\mathrm{opt}2}\leftarrow TSU_{2}$37:     $T_{\mathrm{opt}}\leftarrow T_{2}$38:   **end if**39:**end for**40:Return the optimal network topology $T_{\mathrm{opt}}$


## 5. Simulation and Analysis

### 5.1. Simulation Scenarios

To evaluate the effectiveness and practicality of the proposed topology design algorithm, this study validates the algorithm in the context of mega-constellations in construction or planning.

Our simulation scenarios focus on two typical LEO mega-constellations: the Starlink constellation [44] and the GW constellation [45]. Although the theoretical design of a mega-constellation system may include tens of thousands of satellites, the actual constellation structure typically comprises multiple shells. Owing to differences in orbital altitudes between shells, complex ISLs are generally omitted across shells owing to challenges in engineering implementation and communication efficiency. Therefore, the simulation in this study focuses solely on the topological structure of the Walker constellation within a single shell. Specifically, a typical Starlink shell contains 1296 satellites, while a typical GW shell contains 1728 satellites. The detailed parameters are shown in Table 1. These two Walker constellations exhibit significant differences in orbit parameters, satellite counts, and other aspects, which can provide rich and effective data samples for testing the broad applicability of the proposed algorithm under different Walker–Delta configurations.

The mega-constellation simulation model is constructed using an interconnection between MATLAB R2020b and AGI STK 12.2. The scenario epoch start time is set to 04:00:00.000 UTC on 16 June 2025, with a simulation duration of 1 day (1441 min). This duration is sufficient to cover various dynamic change states during the operation of the satellite constellation, ensuring the integrity and accuracy of the simulation results.

For the setup of ground stations, 200 locations with the largest populations (based on city population data [46]) were selected as ground station sites. The traffic scale between ground stations is generated proportional to the product of the city populations, as specified by Equation (Equation 10).(10)$$f_{ij}=\frac{P_{i}\times P_{j}}{\max(P_{k}^{2})}\;\left(i,j,k=1,2,\ldots ,200\right)\;i\neq j$$
where $P_{i}$ represents the population of city *i*. This formula is designed based on the principle that population sizes reflect communication demand, and thus the product of the populations characterizes the potential traffic correlation between two cities. Normalization by dividing the maximum value of the square of the population of all cities places the traffic values $f_{ij}$ in the range [0,1], providing a unified measurement standard that conforms to the distribution characteristics of actual communication traffic.

For ISL capacity settings, the laser ISL capacity within an orbital plane is $C_{\mathrm{laser}}=10\;\mathrm{Gbps}$, and the radio ISL capacity between different orbital planes is $C_{\mathrm{radio}}=1\;\mathrm{Gbps}$. The traffic matrix is generated using the Python 3.12 NetworkX library, with the total traffic scale controlled at 500 Gbps. We employ the classic Dijkstra algorithm for topological performance verification. By analyzing the path search in the topological snapshots, key statistical data such as transmission delay and hop count are obtained.

### 5.2. Analysis of Simulation Results

In the simulation analysis, a comparative study is conducted among three topology design strategies: Grid-Mesh+, Unified TSU, and Regional TSU. To visually demonstrate the structural differences among these algorithms, Figure 3 provides a schematic illustration of the Grid-Mesh+, Unified TSU, and Regional TSU topologies.

Figure 4a,b present the cumulative distribution function (CDF) of end-to-end delay for 200 cities using the Regional TSU, Uniform TSU, and Grid-Mesh+ topological structure units in the Starlink and GW constellation scenarios, respectively, when λ=1 in the target optimization function (Equation (Equation 4)). The CDF curves show that the delay distributions of all three TSUs exhibit a rapid increase followed by stabilization. Among them, the Regional TSU algorithm demonstrates the optimal delay performance, with its maximum and average delays lower than the other two methods.

Specific data indicates that in the Starlink constellation scenario, the Regional TSU algorithm achieves a 13.25% reduction in average delay compared with Grid-Mesh+, while the Uniform TSU algorithm achieves an 11.65% reduction. The Regional TSU further reduces the average delay by 1.43% compared with the Uniform TSU. In the GW constellation scenario, the average delay of the Regional TSU algorithm is 3.96% lower than Grid-Mesh+, the Uniform TSU is 2.75% lower than Grid-Mesh+, and the Regional TSU is 0.87% lower than the Uniform TSU.

Comparing the two constellation scenarios reveals that the TSU algorithm offers more significant performance improvements in the Starlink constellation. This is because the GW constellation’s orbital altitude (1145 km) is approximately double that of Starlink (550 km), resulting in sparser spatial node distribution and larger ISL spans. Consequently, the potential for delay improvement by adjusting the inter-orbit link connection direction is limited.

Figure 5a,b present the CDF of end-to-end ISL hop count under two constellation scenarios when the weight coefficient λ=1 in the target optimization function (Equation (Equation 4)). The results show that the Regional TSU algorithm again demonstrates optimal performance, with both the maximum hop count and average hop count being lower than those of other TSUs.

In the Starlink scenario, the average hop count of the Regional TSU is reduced by 8.24% compared with Grid-Mesh+, the Unified TSU is reduced by 5.51% compared with Grid-Mesh+, and compared with the Unified TSU, the Regional TSU achieves a further 2.59% reduction. In the GW scenario, the average hop count of the Regional TSU is reduced by 4.80% compared with Grid-Mesh+, the Unified TSU is reduced by 3.58% compared with Grid-Mesh+, and compared with the Unified TSU, the Regional TSU achieves a further 0.98% reduction.

Further analysis of the shared phenomena in Figure 4 and Figure 5 reveals that when the number of ISL hops is small (e.g., the distance between nodes is extremely short, requiring only 1 or 2 hops), the performance differences among the three algorithms are negligible. This is because the path optimization space of the algorithms is inherently limited for short-distance transmissions, demonstrating the performance boundaries of the topology design algorithm in extreme scenarios.

In the preliminary simulations, the weight coefficient λ in the objective function (Equation (Equation 4)) was consistently set to 1. According to the theoretical analysis in Section 2.3 and Section 2.4, fewer hops imply lower network resource consumption and higher capacity. In the operation of mega-constellations, the trade-off between delay and network capacity needs to be considered. Figure 6a,b present the performance comparisons when λ takes values of 0.5, 1, 2, and 5 in the Starlink constellation (results for the GW constellation are similar and thus omitted).

As shown in Figure 6, when λ increases, delay performance improves, but the end-to-end hop count increases, indicating an increase in network resource consumption. Conversely, when λ decreases, the delay performance deteriorates, but the hop count decreases, leading to an increase in network capacity. This phenomenon is consistent with theoretical expectations and shows that operators of mega-constellations can achieve a balance between real-time communication delay requirements and network resource utilization by dynamically adjusting the λ value.

The observed trade-off between delay and hop count, governed by λ, provides mega-constellation operators with a powerful and intuitive knob to tailor the network topology to specific service-level agreements and application profiles. The choice of λ can be guided as follows:High λ values: This configuration prioritizes low latency above all else. It is well-suited for latency-sensitive applications such as real-time interactive services (e.g., online gaming, video conferencing, vehicular networking), where even a few milliseconds of reduction in end-to-end delay can significantly enhance the user experience.Low λ values: This configuration prioritizes network capacity and resource efficiency by minimizing the number of hops. It is ideal for data-heavy, throughput-oriented applications such as bulk data transfer, content distribution, and cloud backup, where the primary objective is to maximize the volume of data delivered, potentially at the expense of higher latency.Intermediate λ values: This configuration seeks a balanced compromise between latency and throughput. It is suitable for general-purpose traffic or scenarios where the traffic mix contains both latency-sensitive and data-heavy components.

In practice, an operator could dynamically adjust λ based on predicted or real-time analyzed network traffic patterns, effectively implementing a policy-driven topology control scheme.

## 6. Conclusions and Future Works

### 6.1. Conclusions

This paper investigated the topology design problem of LEO mega-constellations, addressing the insufficient consideration of the heterogeneous characteristics of ISLs in existing research and the challenges in engineering practice. Aiming to optimize the network performance of mega-constellations, we proposed a heterogeneous ISL architecture that features a “stable high-speed laser backbone connection within intra-orbit planes + dynamic and flexible radio network between inter-orbit planes”. We established a mathematical model and objective function to lay a theoretical foundation for topology design. First, we proved that the topology optimization based on this architecture is an NP-hard problem, confirming that its algorithmic time complexity grows exponentially with the size of the constellation. Then, we defined topological structure units (TSUs) and proposed two topology design algorithms. Finally, through simulation experiments in Starlink and GW constellation scenarios, we verified the effectiveness of the algorithms in improving topological performance, such as reducing end-to-end delay and hop count, and providing new ideas and methods for mega-constellation topology design.

### 6.2. Potential Limitations and Challenges

While the proposed TSU-based topology design offers significant benefits, its implementation and applicability are subject to several limitations:Management Overhead: The dynamic nature of the Regional TSU algorithm, while beneficial for performance, requires periodic (e.g., every ΔT) recalculation and global distribution of the optimal TSUs. This process introduces signaling overhead and computational cost for the network management system, which must be weighed against the performance gains achieved.Orbit Model Specificity: The current algorithm and its analysis are developed and validated within the context of classical single-shell Walker–Delta constellations. The proposed method can be applied to a multi-shell Walker constellation by decomposing it into multiple single shells and applying the topology control within each individual shell. However, the critical challenge of establishing and optimizing inter-shell topological connections between these different orbital layers is not addressed by our current method and represents a significant limitation for deployment in complex, multi-layered constellations.Hardware Implementation Requirements: The proposed heterogeneous architecture mandates that each satellite be equipped with at least two laser inter-satellite links (for intra-orbit connections) and two radio inter-satellite links (for inter-orbit connections).

### 6.3. Future Works

In traffic matrix modeling, for simplified analysis, we only establish its correlation with population density and aggregate multiple ground stations in cities into a single site. This approach simplifies real-world complex scenarios and may affect the accuracy of topology optimization. To address these shortcomings and meet the needs of practical deployment, future research should be expanded in multiple dimensions:This study primarily focused on introducing the TSU concept and evaluating its efficacy through fundamental network performance indicators: latency and hop count. Future work will expand this evaluation framework to encompass a broader set of practical metrics, including the energy consumption of inter-satellite links, the operational complexity of the acquisition, tracking, and pointing (ATP) systems, throughput under congested traffic conditions, robustness to node and link failures (e.g., simulating random ISL outages and measuring network resilience), and the management and signaling overhead associated with the dynamic topology reconfiguration of the Regional TSU algorithm. Investigating the trade-offs between performance gains and these additional costs will be crucial for assessing the overall practicality of TSU-based topology control in next-generation mega-constellations.Investigating influencing factors of traffic matrices, considering business tidal phenomena caused by time zone differences and traffic characteristic variations due to urban functional distributions, to construct more accurate and spatio-temporally dynamic traffic prediction models.Exploring collaborative optimization schemes for topology design and routing algorithms, achieving optimal network resource allocation through joint design, and promoting the development of mega-constellation network technology toward higher efficiency and practicality.

## Figures and Tables

**Figure 1 sensors-25-05840-f001:**
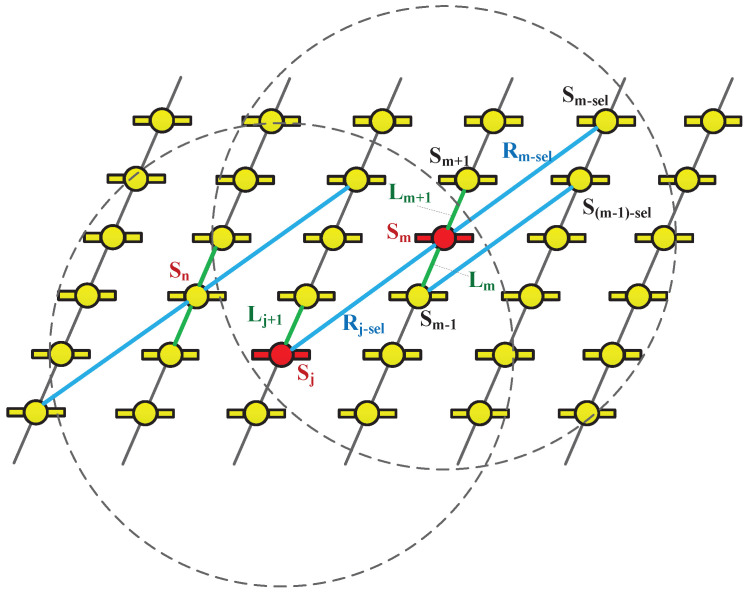
Satellite $s_{m}$ actively establishes intra-orbit laser links (e.g., with $s_{m+1}$, indicated in green) and inter-orbit radio links (e.g., with $s_{m-\mathrm{sel}}$, indicated in blue) within its reachable satellite set (i.e., within a circular area). This core TSU is replicated throughout the constellation, as evidenced by the fact that other satellites adopt an identical link configuration. Ultimately, each satellite establishes four inter-satellite links: two of which are actively established, and the other two are established by other satellites when replicating and applying the TSU.

**Figure 2 sensors-25-05840-f002:**
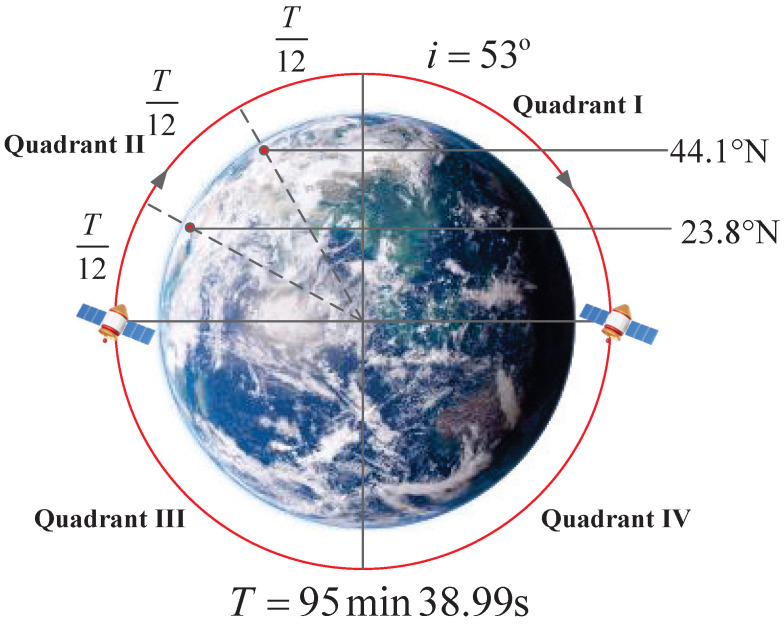
The front projection view of a satellite with an orbital inclination of 53° is divided into four quadrants according to the orbital operation direction.

**Figure 3 sensors-25-05840-f003:**
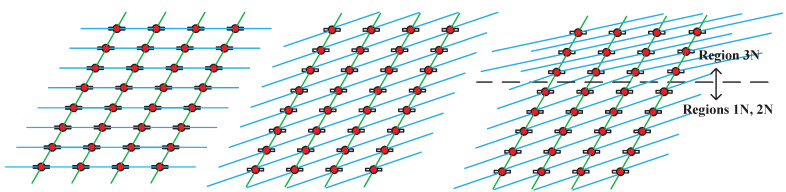
Three topology design strategies (**left**: Grid-Mesh +, **middle**: Unified TSU, **right**: Regional TSU). Satellite nodes are represented by red dots. Intra-orbit laser links are shown in green, and inter-orbit radio links are shown in blue.

**Figure 4 sensors-25-05840-f004:**
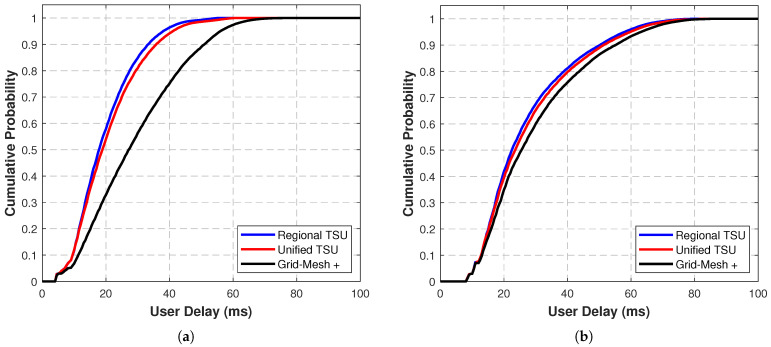
CDF of end-to-end delay for 200 cities using the Regional TSU, Uniform TSU, and Grid-Mesh+ topological structure units: (**a**) in Starlink constellation scenario; (**b**) in GW constellation scenario.

**Figure 5 sensors-25-05840-f005:**
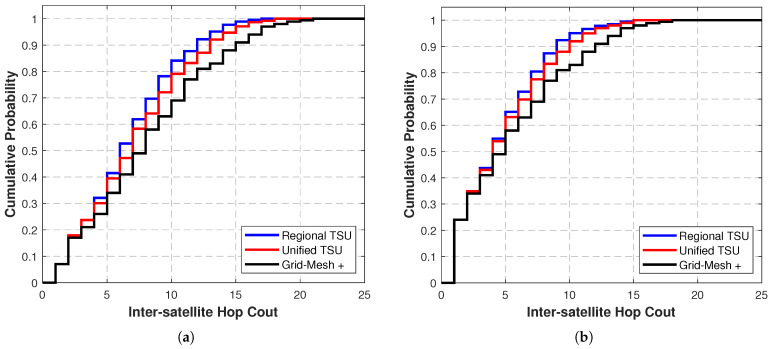
CDF of end-to-end ISL hop count for 200 cities using the Regional TSU, Uniform TSU, and Grid-Mesh+ topological structure units: (**a**) in Starlink constellation scenario; (**b**) in GW constellation scenario.

**Figure 6 sensors-25-05840-f006:**
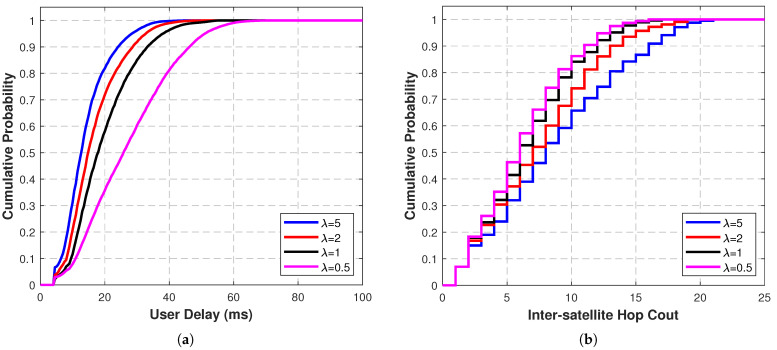
Influence of λ on topology performance in Starlink constellation: (**a**) CDF of end-to-end delay for different λ values; (**b**) CDF of end-to-end hop counts for different λ values.

**Table 1 sensors-25-05840-t001:** Parameters of two typical mega-constellations in simulation scenarios.

Constellation	Number	Altitude (km)	Inclination	Planes	Satellites per Plane	Phase Factor
Starlink	1296	550	53	72	18	45
GW	1728	1145	50	48	36	37

## Data Availability

Data are contained within the article.

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
