# Peer review of "Efficient Topology Design for LEO Mega-Constellation Using Topological Structure Units with Heterogeneous ISLs"

_sensors, 2025, doi:10.3390/s25185840_

Round 1

Reviewer 1 Report

Comments and Suggestions for Authors

Dear authors, here are my observations.

Author Response

Dear Reviewer,

Please download the attached document, which contains our point-by-point responses to your comments.

Reviewer 2 Report

Comments and Suggestions for Authors

Suggested revisions for manuscript ID: sensors-3855587

    This manuscript focuses on the topology design of Low Earth Orbit (LEO) mega-constellations. Aiming at the shortcomings in existing research, such as "prioritizing network layer protocol optimization while downplaying topology structure design" and insufficient consideration of the engineering challenges of inter-orbital Inter-Satellite Links (ISL), it proposes a heterogeneous ISL topology architecture and designs a TSU-based topology algorithm. Finally, simulations are conducted to verify the effectiveness of the algorithm in reducing latency and hop count. The manuscript has certain reference value, but there are still many issues that need to be supplemented or addressed, as follows:

  1. It is recommended to incorporate the "service tidal effect" caused by time zones (e.g., high traffic in cities during the day and low traffic at night) to facilitate the optimization of traffic matrix modeling.
  2. In the manuscript, multiple ground stations are merged into a single endpoint, which may overlook the spatial differences in the actual distribution of ground stations and thus lead to a low degree of matching between topology optimization and actual traffic. It is suggested to consider this factor.
  3. As we know, satellites at high latitudes have a higher relative angular velocity. Will this cause the TSU links to be interrupted at certain times?
  4. What is the basis for selecting the latitude zone divisions of 23.8° and 44.1°? How adaptable is the design to different orbital inclinations, such as i=60° or 53°?
  5. The manuscript only simulates a single-shell Walker constellation. Has any consideration been given to the design of cross-shell ISLs for multi-shell constellations?
  6. Orbital perturbations, such as atmospheric drag, may cause satellite position deviations. What impact does this have on the stability of the long-term topology?
  7. The manuscript considers the "existence conditions" of the ISL model, such as distance and angular velocity, but does not introduce the failure probability of spatial environment interference (e.g., laser links affected by solar radiation and space debris) or the multipath fading of radio links. This makes it difficult to evaluate the fault tolerance of the topology.
  8. Some references lack page numbers or conference information; it is recommended to standardize their formats.

Author Response

(The authors gave the same response as above.)

Reviewer 3 Report

Comments and Suggestions for Authors

This is a good paper with strong potential for publication; however, several issues should be addressed to improve its clarity and overall quality:

- Please expand the related work section, especially contrasting TSU-based design with other modular or hierarchical approaches.

-The justification for the chosen parameters (e.g., the weight factor λ in the optimization function) is not clear. 

-The traffic model used in the simulations is overly simplified. Please clarify the differences between the population-based synthetic traffic model and real-world traffic distributions.

-Consider including additional performance indicators such as throughput under congestion, robustness to ISL failures, or switching overhead.

-Please justify the assumptions made and discuss their practical consequences.

-The claim that the TSU approach is universally applicable appears overstated, given that the evaluation is limited to only two constellations. 

-A comparative schematic explicitly illustrating the differences between Grid-Mesh+, Unified TSU, and Regional TSU topologies would improve the analysis.

Author Response

(The authors gave the same response as above.)

Round 2

Reviewer 1 Report

Comments and Suggestions for Authors

The authors have made the necessary changes and the article looks better. One mention:

- In section 2.2.2 the authors made a satellite link budget sensitivity analysis based on digital modulation schemes. Please also mention the related references.

Author Response

Comment: The authors have made the necessary changes and the article looks better. One mention: In section 2.2.2 the authors made a satellite link budget sensitivity analysis based on digital modulation schemes. Please also mention the related references.

Response: We sincerely thank the reviewer for their positive feedback and for this helpful suggestion. We fully agree that citing relevant literature strengthens the discussion on the sensitivity of the link budget to digital modulation schemes. In the revised manuscript, we have added appropriate references to Section 2.2.2 to support the key points regarding modulation schemes, their spectral efficiency, and their required SNR for a target BER. The added citations are highlighted in red in the revised manuscript. Thank you again for this valuable comment, which has further enhanced the scholarly foundation of our work.

Reviewer 2 Report

Comments and Suggestions for Authors

     All concerns have been basically addressed, and there are no further comments to make. Thank you to the authors for their revisions.

Author Response

Comment: All concerns have been basically addressed, and there are no further comments to make. Thank you to the authors for their revisions.

Response: Thank you very much, and we greatly appreciate your comments.